# The Prototype NOAA Aerosol Reanalysis version 1.0 (pNARA v1.0): Description of the Modeling System and its Evaluation

Shih-Wei Wei[1,2], Mariusz Pagowski[3,4], Arlindo da Silva[5], Cheng-Hsuan (Sarah) Lu[1,2], Bo Huang[3,4]

[1]Joint Center for Satellite Data Assimilation, Boulder, Colorado 80301, US

[2]Atmospheric Sciences Research Center, State University of New York at Albany, Albany, New York 12226, US

[3]Cooperative Institute for Research in Environmental Sciences, University of Colorado Boulder, Boulder, Colorado 80309, US

[4]NOAA Global Systems Laboratory, Boulder, Colorado 80305, US

[5]Global Modeling and Assimilation Office, NASA Goddard Space Flight Center, Greenbelt, Maryland 20771, US

*Correspondence to*: Shih-Wei Wei (swei@albany.edu) and Mariusz Pagowski (Mariusz.Pagowski@noaa.gov)

**Abstract.** In this study, we describe the first prototype version of global aerosol reanalysis at the National Oceanic and Atmospheric Administration (NOAA), the prototype NOAA Aerosol ReAnalysis version 1.0 (pNARA v1.0) that was produced for the year 2016. In pNARA v1.0, the forecast model is an early version of the operational Global Ensemble Forecast System-Aerosols (GEFS-Aerosols) model. The three-dimensional ensemble-variational (3D-EnVar) data assimilation (DA) system

configuration is built using elements of the Joint Effort for Data assimilation Integration (JEDI) framework being developed at the Joint Center for Satellite Data Assimilation (JCSDA). The Neural Network Retrievals (NNR) of Aerosol Optical Depth (AOD) at 550 nm from the MODerate resolution Imaging Spectroradiometer (MODIS) instruments are assimilated to provide reanalysis of aerosol mass mixing ratios. We evaluate pNARA v1.0 against a wide variety of Aerosol Robotic NETwork (AERONET) observations, against National Aeronautics and Space Administration's (NASA) Modern-Era Retrospective

analysis for Research and Applications 2 (MERRA-2; Gelaro et al., 2017; Randles et al., 2017; Buchard et al., 2017) and European Centre for Medium-Range Weather Forecasts' (ECMWF) Copernicus Atmosphere Monitoring Service ReAnalysis (CAMSRA; Inness et al., 2019), and against measurements of surface concentrations of particulate matter 2.5 ($PM_{2.5}$) and aerosol species. Overall, the 3D-EnVar DA system significantly improves AOD simulations compared to observations, but the assimilation has limited impact on chemical composition and size distributions of aerosols. We also identify deficiencies in

the model's representations of aerosol chemistry and their optical properties elucidated from evaluation of pNARA v1.0 against AERONET observations. A comparison of seasonal profiles of aerosol species from pNARA v1.0 with the other two reanalyses exposes significant differences among datasets. These differences reflect uncertainties in simulating aerosols in general.

# 1 Introduction

Aerosols affect Earth's energy balance through the absorption and scattering of solar and terrestrial radiation (Mitchell Jr., 1971). Aerosols also affect the weather and climate through their indirect effects on cloud microphysics, reflectance, and precipitation (Twomey, 1974, 1977; Albrecht, 1989; Jones et al., 1994; Ackerman et al., 2000; Lohmann et al., 2000). In recent decades, the impacts of aerosol direct and indirect radiative effects on numerical weather prediction (NWP) have been studied extensively (Tompkins et al., 2005; Pérez et al., 2006; Mulcahy et al., 2014; Bozzo et al., 2017; Nowottnick et al., 2018).

Aerosol particles can also provide the surface area for deposition of gas phase chemicals, and subsequently affect the sulfuric oxidation and production of new aerosols (Andreae and Crutzen, 1997). The deposition of mineral dust into the ocean affects marine productivity (Chen et al., 2007; Shao et al., 2011). Aerosols also have fundamental impacts on human health by contributing to respiratory, cardiovascular, and allergic diseases (Pöschl, 2005).

In the last decade, several operational weather prediction centers produced global aerosol reanalyses following data
assimilation (DA) methodologies used for meteorological reanalyses. The National Aeronautics and Space Administration (NASA) Global Modeling and Assimilation Office (GMAO) developed an offline (i.e., where the transport is driven using an already reanalyzed dataset) aerosol reanalysis based on the meteorological reanalysis, Modern-Era Retrospective analysis for Research and Applications (MERRA; Buchard et al., 2015). Subsequently, NASA/GMAO produced the version 2 of MERRA (MERRA-2), which provides both aerosol and meteorological reanalyses by assimilating aerosol and meteorological
observations concurrently (Gelaro et al., 2017; Randles et al., 2017; Buchard et al., 2017). The European Centre for Medium-Range Weather Forecasts (ECMWF) produced a global reanalysis of atmospheric composition, including aerosols and trace gasses, known as the Copernicus Atmosphere Monitoring Service (CAMS) interim ReAnalysis (CAMSiRA) (Inness et al., 2013; Flemming et al., 2017), and updated the chemistry and aerosol modules to CAMS reanalysis (CAMSRA) (Inness et al., 2019). The U.S. Navy Research Laboratory (NRL) generated an aerosol reanalysis (Lynch et al., 2016) using NRL Aerosol
Analysis and Prediction System (NAAPS) (Rubin et al., 2016). The Meteorological Research Institute (MRI) of Japan Meteorological Agency (JMA) also produced a global aerosol reanalysis product named the Japanese Reanalysis for Aerosol (JRAero; Yumimoto et al., 2017).

Aerosol reanalysis represents a uniform and continuous best estimate of the true aerosol state in the atmosphere. It is obtained by constraining model forecasts with observations in the process of data assimilation (Lahoz and Schneider, 2014).
Consequently, the reanalysis provides the best available information on temporal and spatial variability of aerosols and can serve as a basis for assessing their impacts on the whole Earth system. For instance, Bozzo et al. (2017) incorporated the 11-year aerosol climatology from CAMSiRA to improve the aerosol direct radiative effect in the global forecast model. Benedetti and Vitart (2018) used CAMSiRA data to initialize the aerosol fields in a prognostic aerosol experiment to assess the impact of direct radiative effect on subseasonal forecasts. Bender et al. (2019) included MERRA-2 as a reference dataset to study the
aerosol-cloud-radiation interaction. Some other studies investigated the regional long-term variability of aerosol activities with one or multiple reanalysis datasets (e.g., MERRA-2 and CAMSRA) (Kalita et al., 2020; Cao et al., 2021; Xian et al., 2022).

Furthermore, aerosol reanalyses can be used as benchmarks in evaluating a newly developed system (Huang et al., 2023 and this study).

The model is a key component in production of aerosol reanalysis. For the consideration of computational constraints, complex chemistry involving aerosols needs to be reduced by employing parameterizations which only capture the most essential processes. For instance, the Goddard Chemistry Aerosol Radiation and Transport (GOCART) utilized in MERRA-2 assumes no interaction between aerosol species (i.e., external mixing) and considers sulfur oxidation with prescribed climatology for OH, $NO_3$, and $H_2O_2$ (Chin et al., 2000, 2002; Colarco et al., 2010). In contrast, CAMSRA (Inness et al., 2019) considers a more comprehensive chemistry, such as the parameterization of the secondary organic aerosols (not considered in CAMSiRA). In NAAPS, the primary and secondary organic aerosols are preprocessed at the initialization stage. The Model of Aerosol Species IN the Global AtmospheRe (MASINGAR mk-2) which is used to produce JRAero also simplifies the processes involving production of secondary organic aerosols. Besides the simplification of chemistry, the interaction between aerosols and atmospheric physics is usually neglected. Both MERRA-2 and CAMSRA radiatively coupled the prognostic aerosols but did not consider the cloud-aerosol interaction (Gelaro et al., 2017; Inness et al., 2019). The climate model of JRAero actively coupled the aerosol information from MASINGAR mk-2 to the radiation and the two-moment bulk cloud microphysics schemes (Yukimoto et al., 2012).

Assimilated observations are a critical component of a reanalysis product. For aerosols, these are typically retrievals of Aerosol Optical Depth (AOD) at 550 nm. Because AOD is a measure of extinction of light over a whole atmospheric column, it carries a limited amount of information on the chemical composition and sizes of particles and also cannot provide information on the vertical distribution in the atmosphere. Consequently, the resulting analysis is largely determined by the prior information contained in the model state at a given time. This provides very limited ability for AOD DA to correct aerosol speciation, size, and vertical distributions in the atmosphere.

In 2011, the National Oceanic and Atmospheric Administration (NOAA) National Centers for Environmental Prediction (NCEP) implemented the NOAA Environmental Modeling System (NEMS) Global Forecast System (GFS) Aerosol Component, version 1 (NGAC v1; Lu et al., 2016) with GOCART parameterization for dust-only forecasts. Later, NCEP upgraded the system to NGAC v2, which provided forecasts for all aerosol species in GOCART parameterization (Wang et al., 2018; Bhattacharjee et al., 2018). In September 2020, NGAC was replaced by the Global Ensemble Forecast System (GEFS)-Aerosols model which relies on the Finite-Volume Cubed-Sphere (FV3) dynamical core and GOCART parameterization (Zhang et al., 2022). Currently, the aerosol forecasting system at NCEP does not include DA, despite previous studies showing significant impact of AOD observations on aerosol initial conditions and the subsequent forecasts (Benedetti et al., 2009; Pagowski et al., 2010, 2014; Schwartz et al., 2014). A novel three-dimensional ensemble-variational (3D-EnVar) DA system based on elements of the Joint Effort for Data assimilation Integration (JEDI) has been developed at NOAA to provide the best estimates of atmospheric aerosol distributions by assimilating AOD at 550 nm from Visible Infrared Imaging Radiometer Suite (VIIRS) (Huang et al., 2023).

In this study, we used and designed a specific JEDI-based 3D-EnVar DA configuration to produce the NOAA Aerosol ReAnalysis version 1.0 (NARA v1.0). The reanalysis for year 2016 was produced as a pilot product and its evaluation is presented here. We evaluated the year-long reanalysis against independent observations and the gridded aerosol products from MERRA-2 and CAMSRA. This paper is structured as follows. Section 2 describes the observations used in this study. Sections 3 and 4 describe the forward model and the DA system, respectively. Section 5 reports results of evaluations. Section 6 concludes this study and outlines the future development.

## 2 Observations

NARA v1.0 assimilates the Neural Network Retrieval (NNR) AOD derived from observations of the MODerate resolution Imaging Spectroradiometer (MODIS) onboard Terra and Aqua satellites (Castellanos and da Silva 2017; Randles et al., 2017). Unlike the standard retrieval method that relies on a physical model (e.g., a radiative transfer model), the NNR approach retrieves the AOD based on a well-trained model conducted by a neural network machine learning method. The training dataset contains MODIS level 2 reflectances, glint, solar and sensor angles, cloud fraction, and albedo derived using GEOS-5 surface wind speeds or climatology over ocean or land, respectively, targeting AOD observations from Aerosol Robotic NETwork (AERONET). The NNR AOD is available at the following wavelengths: over ocean: 470, 500, 550, 660, and 870 nm; over dark land: 440, 470, 500, 550, 660, 870 nm; and over bright land: 440, 470, 500, 550, 660 nm. Only MODIS NNR AOD at 550 nm has been assimilated in MERRA-2 (Randles et al., 2017) and pNARA v1.0 (this study).

For evaluation of pNARA v1.0 and the free model run, we used the version 3 level 2.0 AOD retrievals from AERONET (Giles et al., 2019), particular matter 2.5 ($PM_{2.5}$) measurements from OpenAQ dataset (https://openaq.org, last access on 03 May 2023), and speciated aerosol surface concentration measurements from the Interagency Monitoring of Protected Visual Environments (IMPROVE) network (Hand et al., 2019, http://vista.cira.colostate.edu/Improve/improve-program/, last access on 03 May 2023). AERONET is a ground-based remote sensing aerosol network, which uses a Sun photometer to measure direct solar irradiance and provides the AOD retrievals at 340, 380, 440, 500, 670, 870, 940 and 1020 nm. In the past more than 25 years, the network has expanded to more than 600 stations. The manual quality control used in version 2 leads to a significant delay in the level 2.0 database. AERONET version 3 fully automated the quality control algorithm to screen out cloud contaminated data, which eliminates the manual efforts and reduces the processing time for quality-assured data (i.e., level 2.0).

OpenAQ is an open source platform collecting air quality measurements from various sources globally. In this study, we only use governments/research insititutes maintained (i.e., 'reference grade') stations, which are majorly over US and Europe, to evaluate the performance of pNARA v1.0. Besides $PM_{2.5}$ measurements, OpenAQ database also ingests the measurements of gas phase pollutants, such as $O_3$, $NO_2$, and $SO_2$, which can be utilized as an independent dataset for trace gases evaluation in future studies. The IMPROVE network was established as the visibility network over US in 1985. It derives visibility metrics

through the measurement of speciated aerosol mass concentrations, including anions of sulfate, nitrate, nitrite, chloride, and organic and elemental carbonaceous species.

## 3 Model description

In pNARA v1.0, we use an early Common Community Physics Package (CCPP) version of the operational Global Ensemble Forecast System-Aerosols (GEFS-Aerosols) model. GEFS-Aerosols provides prognostic mass mixing ratios of five bins dust, five bins sea salt, hydrophobic and hydrophilic black and organic carbon, and sulfate. It relies on GOCART parameterization implemented with the GFS physics package and Finite-Volume Cubed-Sphere (FV3) dynamical core (Lin, 2004; Lin et al., 2017). It handles sub-grid transport and wet scavenging of aerosols in the atmospheric physics module. Other chemical processes, such as emissions, chemical reactions, dry deposition, and settling, are handled by the chemical module driven by meteorological fields from the atmospheric component of the model. Note that the formation of secondary organic aerosols and nitrates are not considered in the GEFS-Aerosols. The model has been recently updated for operations to reduce some biases (Zhang et al., 2022), but the most recent version was not available at the time of the execution of this study.

In GEFS-Aerosols, the background fields of OH, $H_2O_2$, and $NO_3$ used in the parameterization of simplified sulfur chemistry in GOCART are updated with a monthly mean climatology from the 2015 version of NASA Global Modeling Initiative's (GMI) chemical model. The model also includes the 1D plume rise module adapted from the High-Resolution Rapid Refresh (HRRR) Smoke model to improve vertical distribution of smoke emissions. The biomass burning emissions use the version 3 of Blended Global Biomass Burning Emissions Product (GBBEPx v3; Zhang et al., 2019). This parameterization blends the Quick Fire Emissions Dataset (QFED) used in MERRA-2 (Darmenov and da Silva, 2015) and daily emissions derived by the hot spots and the fire radiative power observations from polar-orbiting and geostationary satellites. The anthropogenic emissions are based on the inventories from Community Emissions Data System (CEDS). The sea salt scheme has been updated to the recent version of GOCART scheme (Colarco et al., 2010). The dust scheme has been updated to FENGSHA (Tong et al., 2016; Zhang et al., 2022), which means the 'wind-blown dust' in Mandarin Chinese.

## 4 Data Assimilation system

To produce pNARA v1.0, we used a three-dimensional ensemble-variational (3D-EnVar) aerosol data assimilation system using components from the Joint Effort for Data assimilation Integration (JEDI; Huang et al., 2023). JEDI is primarily developed at the Joint Center for Satellite Data Assimilation (JCSDA) but receives significant contributions from partner and sponsor agencies (i.e., NOAA, NASA, US-Navy, US-AirForce, and UK MetOffice). It aims to provide an integrated and unified DA framework for Earth system prediction applications and reduce redundant efforts across research and operational communities. More information about JEDI can be found at the JCSDA website (https://www.jcsda.org/jcsda-project-jedi, last access on 03 May 2023).

The forward observation operator that converts model variables (mass mixing ratios of aerosols and pressure layer depths) to AOD observations relies on aerosol specific extinction coefficients interpolated from look-up tables (i.e., 'AodLUTs' operator in JEDI). Tabulated values were obtained at NASA/GMAO from Mie theory for spherical particles (Wiscombe, 1980) and from the T-matrix approach for non-spherical dust (Meng et al., 2010).

In this application, the 3D-EnVar aerosol DA system assimilates MODIS NNR AOD retrievals at 550 nm to the GEFS-Aerosol model every 6 hours. For MODIS NNR AOD, it was screened the cloud-affecterd data and trained with AERONET measurments during the neural network training (Randles et al., 2017). We also thinned the horizontal resolution from 10 km to 50 km to reduce the correlation between observations. The result is a reanalysis of mass mixing ratios of five GOCART aerosol species, with sea salt and dust distributed over five size bins. Figure 1 displays the schematic of the 3D-EnVar aerosol DA system. The 3D-EnVar system used for pNARA v1.0 is a combination of a Local Ensemble Transform Kalman Filter (LETKF; Bishop et al., 2001; Hunt et al., 2007) using 40 members and a 3D-Variational (3D-Var; Lorenc, 1986) solver that produces the control analysis using the ensemble information to obtain the background error covariance matrix. The LETKF analyses are recentered on the 3D-Var analysis to ensure consistency between the two analyses (Hamill and Snyder 2000; Lorenc 2003; Buehner 2005). The control analysis constitutes the reanalysis output. The forecasts and analyses of control and ensemble members are conducted at C96 (~100km) resolution. Given the coarse resolution, this study focus on the larger temporal and spatial scale performance. Note that no other observations are assimilated since meteorological fields in the control and in the ensemble are driven by analyses from the Global Data Assimilation System (GDAS) at NCEP.

To address model biases and spread deficiency of the ensemble, a scheme to scale and perturb source emissions was devised. Scaling factors are derived for dominant aerosols based on AOD deficits over regions. The scale factor of 2 is selected for all aerosol species to increase the emission rate for reducing the model biases. Perturbations to emissions of ensemble members represent spatially and temporally correlated patterns following the approach in the Stochastically Perturbed Parametrization Tendencies (SPPT; Palmer et al., 2019) scheme. With applying SPPT to emissions, it can enhance the variance between ensemble members (i.e., larger ensemble spread) thus the influences of observations to the analysis are increased. This approach and a method to obtain scaling factors and amplitudes of emission perturbations are detailed in Huang et al. (2023).

## 5. Evaluation

In this section, we divide the year of 2016 into four seasons to investigate the performance of pNARA v1.0. Winter includes December, January, and February (denoted as DJF); spring includes March, April, and May (denoted as MAM); summer includes June, July, and August (denoted as JJA); and fall includes September, October, and November (denoted as SON).

### 5.1 Comparison against AERONET

Figure 2 illustrates the global seasonal comparison of AOD at 500 nm from pNARA v1.0 and the free model run with respect to AERONET and the statistics are also provided. This table lists absolute and relative biases, $R^2$-correlations (also known as

coefficients of determination) and, for reader's convenience, correlation coefficients (R) for each season and the whole year for the free model run and pNARA v1.0. It is followed by Fig. 3 where time series displaying the bias and the $R^2$-correlation scores. Because the standard AOD at 550 nm is not directly observed in AERONET, we chose to perform the evaluation of AOD at 500 nm to avoid interpolation and log-linearization errors in AERONET observations. The free model run considerably and systematically underestimates the AOD throughout the year. Among four seasons, the largest bias between model and measurements occurs in winter (i.e., DJF). Compared to the free model run, our reanalysis, pNARA v1.0, has a substantially better agreement with AERONET throughout the whole year in terms of bias and correlation. We note that the statistical scores are varying seasonally as they are influenced by the density of AERONET observations that are used in the evaluation (the highest/lowest density of observations occurs in the summer/winter over North America and Europe).

As can be seen in Fig. 3, the performance of the free model run in terms of bias is the best during summer months and the worst from January to March. For pNARA v1.0, the negative absolute biases are much reduced (on average by about 0.05) though not enough to remove the overall negative bias of the reanalysis. pNARA v1.0 also has significantly improved $R^2$-correlation throughout the whole year (by about 0.2). Statistics of the free model run and reanalysis vary throughout the year nearly in parallel demonstrating the crucial role of the forecast model in data assimilation. In other words, data assimilation cannot drastically improve the quality of simulations if the forecast model is seriously deficient.

Figure 4 displays the probability density plots of 440-870 nm Ångström Exponent (AE) for the free model run and the reanalysis against the AERONET observations throughout 2016. AE is calculated by the Equation 1 below, where $\lambda_1$ and $\lambda_2$ are 440 and 870 nm, respectively. It provides a measure of relative extinction of light by aerosols at different wavelengths and primarily reflects the size distribution of particles (Schuster et al., 2006).

$$AE(\lambda_1, \lambda_2) = -\frac{log(AOD(\lambda_1)/AOD(\lambda_2))}{log(\lambda_1/\lambda_2)}, \tag{1}$$

In general, the differences between the free model run and the reanalysis are insubstantial, and both correlate poorly with observations. The lack of an improvement in the reanalysis demonstrates that assimilating AOD at 550 nm alone only minimally impacts size distribution and/or composition of aerosols. Saide et al. (2013) and Tsikerdekis et al. (2021) have shown that a more realistic representation of aerosols can be produced by assimilating multi-wavelength retrievals of AOD, fine-mode fraction AOD and single scattering albedo. We hope more aerosol retrieval products from the future missions, such as Plankton, Aerosol, Cloud, ocean Ecosystem (PACE) mission (https://pace.gsfc.nasa.gov, last access on 03 May 2023), can introduce more information into the analysis system in the near future.

In Fig. 5, probability density plots of 440-675 nm Absorption AE (AAE) versus Scattering AE (SAE) are matched for AERONET Almucantar retrievals (Sinyuk et al., 2020) and pNARA v1.0. For brevity, a similar scatter plot for the free model run is omitted since it is only marginally different from the latter. We obtained model's AAE and SAE with Equations 2a and 2b,

$$AAE(\lambda_1, \lambda_2) = -\frac{log(AAOD(\lambda_1)/AAOD(\lambda_2))}{log(\lambda_1/\lambda_2)}, \tag{2a}$$

$$SAE(\lambda_1, \lambda_2) = -\frac{log(SAOD(\lambda_1)/SAOD(\lambda_2))}{log(\lambda_1/\lambda_2)}, \tag{2b}$$

where absorption AOD (AAOD) and scattering AOD (SAOD) at wavelengths $\lambda_1$ (equal to 440 nm) and $\lambda_2$ (equal to 675 nm) are calculated using the Equations 3a and 3b. The calculations use single scattering albedo (SSA) and extinction coefficient (Ext) at corresponding wavelengths ($\lambda$), mixing ratios (q) for each aerosol species (s), dry air density ($\rho$), and layer thickness (dz) for each pressure level (p).

$$AAOD(\lambda) = \sum_p \quad [\sum_s \quad (1 - SSA_{s,\lambda}) \times Ext_{s,\lambda} \times q_{s,p}] \times \rho(p) \times dz(p) \tag{3a}$$

$$SAOD(\lambda) = \sum_p \quad [\sum_s \quad SSA_{s,\lambda} \times Ext_{s,\lambda} \times q_{s,p}] \times \rho(p) \times dz(p) \tag{3b}$$

Based on Cazorla et al. (2013), the whole plane in the left panel of Fig. 5 was further divided into separate sections representing different types of dominant aerosol species. These authors associate AAE with the representation of chemical composition and SAE with the representation of particle sizes. Figure 5 reveals important deficiencies in the representation of optical properties of aerosols in the model. Because of the dependence of absorption on refractive indices of aerosols, we believe that the strong

lack of variability in AAE (vertical axis in Fig. 5) in pNARA v1.0 compared to AERONET is explained by assumptions and simplifications in GOCART parameterization. These are, in the order of importance: the external mixing of the aerosols in the model, uniform mineralogy of dust sources across the globe, unaccounted variety of organic aerosols and prescribed size distributions of particles. A separate investigation would be required to ascertain the impacts of the above factors on the realism of simulations. Given the importance of radiation absorption by atmospheric aerosols in stratification and clouds, our exposure

of the GOCART shortcomings identified above highlight uncertainties in modeling meteorology-chemistry interactions in weather and climate models that rely on this or similar schemes.

## 5.2 Comparison against MERRA-2 and CAMSRA reanalyses

In this section, the AOD and aerosol vertical distribution in pNARA v1.0 are evaluated against MERRA-2 and CAMSRA. To do the direct comparison between these three datasets, we post-processed the aerosol mass mixing ratio in pNARA v1.0 and

MERRA-2 from its model levels to the same pressure levels as CAMSRA based on the level thickness and surface pressure. For any levels below the terrain, the surface value is assigned, which is the same approach as in CAMSRA.

Figure 6 shows the seasonal mean biases of AOD at 550 nm of pNARA v1.0 with respect to CAMSRA and MERRA-2. Overall, pNARA v1.0 is closer to MERRA-2. This can be attributed to the similarity of aerosol modules (i.e., GOCART) and the same FV3 dynamical cores (Lin, 2004; Lin et al., 2017) in the models used in NOAA and NASA reanalyses. However,

pNARA v1.0 generates substantially smaller dust plumes over Sahara overall and especially during summer. Compared to CAMSRA, pNARA v1.0 shows significant discrepancies over oceans throughout 2016. It also has lower aerosol concentrations over India, especially during the wintertime. The AOD in pNARA v1.0 is usually lower over eastern China with the largest differences occurring in the spring. These instances of lower aerosol concentrations could be attributed to the fundamental differences of aerosol modules between pNARA v1.0 and CAMSRA. Throughout 2016 pNARA v1.0 generated more dust

over Gobi Desert compared to both CAMSRA and MERRA-2. Occasionally, concentrations of dust over this area in pNARA

v1.0 seem unrealistically high. This suggests the need for further improvements to the model parameterization of dust uplift, which is determined by meteorological and land surface conditions.

While the size distributions of dust and sea salt are identical in MERRA-2 and pNARA v1.0, they are different from those in CAMSRA. Approximate conversions between the models would be possible but not straightforward since the bin sizes and distribution parameters for these two aerosol types differ quite significantly (e.g. dust bins in MERRA-2: [0.1, 1.0, 1.8, 3.0, 6.0, 10.0] µm, in CAMSRA: [0.03, 0.55, 0.9, 20.] µm, sea salt bins in MERRA-2: [0.03, 0.1, 0.5, 1.5, 5.0, 10.0] µm in dry conditions, CAMSRA: [0.03, 0.5, 5., 20.] µm at relative humidity 80%). Therefore, in the following we chose to compare total mass mixing ratios of these aerosol species for all the reanalyses. Comparison of vertical profiles of various aerosols was performed over several geographic areas where different aerosols are expected to dominate (Fig. 7). Here, for illustration purposes only, we compare the dust aerosols over North Africa and Middle East (NAFRME) and North Atlantic Ocean (NATL; transported dust), the carbonaceous aerosols over Equatorial and South Africa and the surrounding tropical Ocean (SAFRTROP) and Siberia (RUSC2S), the anthropogenic aerosols over East Asia (EASIA), and the sea salt aerosols over Southern Ocean (SOCEAN).

Figure 8 displays the vertical profiles of total dust mass mixing ratios from the free model run, pNARA v1.0, MERRA-2, and CAMSRA over NAFRME and NATL during the summer. In general, the four datasets show variable discrepancies in dust depending on the region of interest. Over the NATL region the dust profiles are similar to each other while over the NAFRME region GOCART-based models generate higher dust mass mixing ratios near the surface (in particular with the MERAA-2 dataset). Where the Saharan dust are uplifted over the NAFRME region the free model run creates a more vertically mixed profile while pNARA v1.0 and MERRA-2 show higher values near the surface. By assimilating the similar observation dataset (i.e., MODIS NNR), the 3D-EnVar system corrects the analysis toward MERRA-2.

Figure 9 shows a comparison of profiles of carbonaceous aerosols over SAFRTROP and RUSC2S during JJA. As illustrated in Fig. 6, high values of AOD systematically occur over these areas in 2016. For CAMSRA, concentrations of all carbonaceous aerosols are significantly higher near the surface when compared to the other simulations. Interestingly, much higher concentrations of hydrophilic organic and black carbons (OC and BC) exist for CAMSRA at mid-levels. A comparison between the free model run and pNARA v1.0 shows that DA introduced more carbonaceous aerosols over RUSC2S, while there are no considerable differences over SAFRTROP. Comparison of all these simulations indicates that there exist notable discrepancies between vertical profiles of carbonaceous aerosols over areas where extensive wildfires occurred. These discrepancies can be attributed to different parameterizations of biomass burning emissions in the models. For instance, GEFS-Aerosols utilized the GBBEPx biomass burning emissions (Zhang et al., 2019), which is conducted based on QFED (Darmenov and da Silva, 2015) and the fire radiative power observations from polar-orbiting and geostationary satellites. CAMSRA used biomass burning emissions from Global Fire Assimilation System (GFAS; Kaiser et al., 2012), which shows lower emission compared to QFED (Pan et al., 2020). Also, the smoke plume rise is considered in GEFS-Aerosols but not in MERRA-2 and CAMSRA. These attest to large uncertainties that exist in parameterizing wildfires.

Figure 10 illustrates vertical profiles of sulfate and dust aerosols over East Asia (EASIA) during DJF and MAM. The results are consistent with the negative AOD biases of pNARA v1.0 over EASIA shown in Fig. 6. AOD deficit over EASIA in pNARA v1.0 compared to the other reanalyses occurs as a consequence of considerably lower concentrations of both sulfate and dust aerosols.

In Fig. 6, pNARA v1.0 displays significant negative AOD bias over oceans with respect to CAMSRA and smaller bias over oceans with respect to MERRA-2. We chose the area marked as SOCEAN in Fig. 7 to investigate reasons for the differences between the reanalyses. In Fig. 11, we show vertical profiles of mixing ratios of selected aerosol species over this area during JJA and SON. Among the three reanalyses sea salt aerosols were lifted to higher elevation in CAMSRA resulting in larger AOD values over the oceans. AOD assimilation in pNARA v1.0 leads to a better agreement with CAMSRA and MERRA-2 compared to the free model run, but the sea salt loading is still lower during JJA. It is worth mentioning there exist divergences between reanalyses in profiles of mass mixing ratios of other aerosol species though the values are small. MERRA-2 has the most dust while CAMSRA has the most hydrophobic OC. Both MERRA-2 and CAMSRA have more sulfate aerosols than pNARA v1.0. Shapes of vertical profiles of mass mixing ratios of dust and sulfate vary widely between the reanalyses.

In conclusion, we note that assimilation of AOD leads to convergence of pNARA v1.0 towards both MERRA-2 and CAMSRA. This statement holds for AOD and generally for vertical profiles of aerosols. However, assimilation of an integrated quantity such as AOD over a single wavelength (here 550 nm) has little impact on the "shape" of vertical profiles and values of mass mixing ratios usually appear to be scaled proportionally throughout the depth of the atmosphere. Most importantly, there exist marked differences in seasonal vertical profiles of aerosol species between the three reanalyses.

### 5.3 Comparison against in-situ measurements of surface aerosol concentrations

In the following we present an evaluation of pNARA v1.0 against measurements of surface $PM_{2.5}$ concentrations from OpenAQ and concentrations of aerosol species from IMPROVE.

Hourly measurements of surface concentrations of $PM_{2.5}$ collected in the global OpenAQ database are available from late April 2016 onwards. First, we note that in-situ measurements display high spatial and temporal variability, which reflects the origin and subsequent evolution of this species. The horizontal resolution of our model (about 100 km), which also affects accuracy of the representation of terrain topography, is far too coarse for the results to be compatible with such measurements. Also, these measurements are obtained close to the ground that is significantly lower than the bottom level of our model (on average about 20 m). This makes the calculation of the concentration at the surface dependent on the atmospheric stratification in the surface layer. Because of uncertainties involved, here, we simply multiply the mixing ratio of model $PM_{2.5}$ expressed in Equation 4 by moist air density obtained from the model diagnostic 2-meter temperature and humidity and surface pressure.

$$PM_{2.5} = BC1 + BC2 + OC1 + OC2 + SULF + DUST1 + DUST2 \times 0.38 + SEAS1 + SEAS2 + SEAS3 \times 0.83, \quad (4)$$

In this equation acronyms denote the following aerosols species: BC - hydrophilic and hydrophobic black carbonaceous, OC - hydrophilic and hydrophobic organic carbonaceous, SULF - sulfate, DUST - dust in two smallest size bins, and SEAS - sea salt in three smallest size bins. Finally, we note that AOD which represents a column-integrated aerosol quantity may poorly

correlate with surface values alone since aerosol-rich layers often occur at raised elevations. Nevertheless, in our opinion, such evaluation statistics not only provide valuable information on the reliability of global model results for $PM_{2.5}$ forecasting and analysis but also on relevance of assimilating AOD for such purposes. The statistics were calculated for 00, 06, 12, and 18 UTC and are presented on probability density plots in Fig. 12. Model bias with respect to in-situ $PM_{2.5}$ measurements coincides with its bias against in-situ AERONET AOD retrievals; the correlation with $PM_{2.5}$ measurements is much poorer compared to the correlation with AERONET AOD retrievals. Overall and modest improvements in statistics can be noted for the reanalysis compared to the model free run. The spatial density of OpenAQ measurements varies considerably over the globe and it is the highest over North America, Europe, and East Asia. To account for the geographical variability, we performed tests with spatial thinning of data but sensitivity of the statistics to this procedure was minor.

Measurements of daily averaged (beginning at midnight local standard time) concentrations of selected aerosol species by the IMPROVE network occur once in three days and are limited to North America and a single site in South Korea. The sites are located remote from industrial and population centers. For comparison with the model following guidance from Hand at al. (2019) measured concentrations of organic carbonaceous species and sulfate were scaled by 1.8 and by 1.375, respectively. Six-hourly model mixing ratios of species were weighted appropriately to account for local time of the measurements and, as for comparison with $PM_{2.5}$ from OpenAQ, multiplied by 2-meter moist air density. Probability density plots for the carbonaceous species, sea salt, and sulfate for the reanalysis, which differ little from the free model run, appear in Fig. 13. Poor performance of the model and lack of improvement from the assimilation can be noted. Because of the scarcity of systematic measurements of concentrations of individual aerosol species, or lack thereof, outside of North America our statistics only reflect performance of the model over this geographical area. It would be, however, overly optimistic to expect that the model skill would significantly improve elsewhere.

## 6. Conclusions

This study documents the development and the evaluation of pNARA v1.0, which is the first global aerosol reanalysis product at NOAA. pNARA v1.0 is made available for distribution at https://esrl.noaa.gov/gsd/thredds/catalog/retro/global_aerosol_reanalysis/catalog.html (last access on 03 May 2023). To produce pNARA v1.0, the GEFS-Aerosols model was used to forecast aerosol mass mixing ratios. The GEFS-Aerosols relies on GOCART parameterization coupled with FV3 dynamical core using GFS physics. The reanalysis containing 3D aerosol mass mixing ratios for 2016 was generated in a process of a 3D-EnVar data assimilation using specifically designed JEDI configuration. The system assimilated Neural Network Retrievals of AOD at 550 nm from MODIS instruments onboard Terra and Aqua satellites. We evaluated the AOD against AERONET observations, the reanalyses from NASA GMAO's MERRA-2 and ECMWF's CAMSRA, and measurements of surface concentrations of $PM_{2.5}$ and selected aerosol species.

The evaluation against AERONET observations shows that the assimilation of AOD retrievals at 550 nm improves the overall agreement with AOD at 500 nm, especially in the spring and summer. However, the comparison of Ångström Exponent (AE)

indicates that the assimilation of the single-wavelength AOD induces minimal improvements to the speciation and the size distributions of aerosols. Furthermore, the significant underdispersion in the probability density plot of Absorption AE vs. Scattering AE along the ordinate (Fig. 5) indicates, what we believe, is a shortcoming of the GOCART parameterization. We suspect that this shortcoming stems primarily from the assumption of the external mixing of aerosols (i.e., no interaction among aerosol species) in this scheme.

In terms of AOD at 550 nm, pNARA v1.0 shows close proximity to MERRA-2 while it has significant negative biases against CAMSRA. pNARA v1.0 shows lower amounts of dust aerosol over the Sahara and Middle East. The GEFS-Aerosols model and consequently our reanalysis display higher dust concentrations over Gobi Desert than both reanalyses. These deficiencies over deserts call for improvement to model parameterization of dust uplift. For biomass burning areas, GOCART-based reanalyses, MERRA-2 and pNARA v1.0, share similar characteristics while CAMSRA shows significantly different amounts and ratios of carbonaceous aerosol species. Over East Asia, pNARA v1.0 displays significantly lower amounts of aerosols than CAMSRA and MERRA-2. This is due to the lower concentrations of sulfate aerosols throughout the year and dust during winter and spring. The discrepancies between the reanalyses indicate that there exist significant differences in parameterizations of anthropogenic, biomass burning and wind-driven dust uplift between the three models.

In this study, we demonstrated that the reanalyses of AOD at 550 nm produced at NASA (Buchard et al., 2017), ECMWF (Inness et al., 2019), and NOAA (this study) show marked differences globally (see Table A1 in Appendix A for further comparison among reanalyses). Also, seasonally averaged vertical profiles of aerosol species vary significantly between the three reanalyses. This pertains both to the chemical composition of aerosol mixtures and the shapes of the vertical profiles of species. The differences between the reanalyses may arise from distinct model numerics, physical and chemical parameterizations, boundary conditions as sources of tracers, and approaches in data assimilation. These observations combined with the poor representation of the absorption spectrum (Fig. 5) by simple aerosol schemes such as GOCART leads us to a conclusion that our assessment of the state of atmospheric aerosols and their radiative impacts is hardly adequate to allow detailed forecasts of stratification and clouds with aerosol-sensitive physical parameterizations. For instance, Bozzo et al. (2017) and Mulcahy et al. (2014) have demonstrated the importance of including more realistic aerosol states in the parameterization of aerosol-cloud-radiation interactions. For reasons noted above, we believe that at the current state of aerosol science large uncertainties in simulating aerosol-meteorology interactions in the weather and climate models exist.

Comparisons with measurements of surface concentrations of $PM_{2.5}$ from OpenAQ show limited skill of the model and reanalysis in this task (see Table A2 in Appendix A for further comparison among reanalyses). Performance of the model and reanalysis was poor when the results were compared with surface concentrations of carbonaceous, sulfate, and sea salt aerosols. Given the limitations of the model that were listed in the section dedicated to its evaluation against measured concentrations of aerosols at the surface, such results may not be unexpected but are nevertheless disappointing since they suggest that our global reanalysis has a limited value for those health and epidemiological studies in which chemical composition of aerosols is considered. It is possible that extending chemical parameterizations to include the formation of secondary organic aerosols would improve model forecasts (Fan et al., 2022).

Future space missions (e.g., PACE that we mentioned above) and new algorithms (e.g., Zhou et al., 2021) promise to provide novel retrievals of multi-wavelength solar and lunar AOD, fine-fraction AOD, single scattering albedo, and retrievals of aerosol layer height. These developments should enhance scope of evaluations, pose additional constraints in data assimilation, and eventually lead to better aerosol forecasts and reanalyses.

The main goal of this manuscript is to present our initiative to produce the first-ever global aerosol reanalysis at NOAA. As discussed above, our reanalysis has deficiencies that will be addressed in turn. Huang et al. (2023) outlined the potential enhancements to our assimilation approach, estimates of observation and model errors, and systemic correction of model biases. We will report our advances in the future.

## Appendix A

To satisfy the reviewer's request, we provide the spatial distribution of temporal statistics from experiments (Fig. A1 and A2) and the comparison of the three reanalysis datasets (Table A1 and A2).

Figure A1 and A2 display the temporal biases, $R^2$, and RMSE of AOD at 500 nm against the AERONET retrievals (Fig. A1) and the $PM_{2.5}$ measurements against OpenAQ (Fig. A2) from both free model run and pNARA v1.0 analyses. For AOD at 500 nm, pNARA v1.0 shows smaller biases, better correlation, and reduced RMSE in most of places. However, over Asia (e.g., India and Southeast Asia) regions, the improved correlation and similar RMSE indicate the reanalyses from pNARA v1.0 provide the better temporal variation but not the magnitude. In terms of the $PM_{2.5}$, Figure A2 shows no discernible differences between two experiments. It indicates the DA system with AOD retrievals barely helps the PM simulations in GEFS-Aerosols. Note that stations over India are substantially underestimated compared to the stations over US and Europe. It suggests the $PM_{2.5}$ in this area involves more complicated chemical processes, which is not resolved by the simple chemistry in the GOCART model.

Table A1 and A2 show the station averaged temporal statistics for biases, RMSE and $R^2$ from pNARA v1.0, CAMSRA, and MERRA-2 with respect to AOD at 550 nm (Table A1) from AERONET and $PM_{2.5}$ from OpenAQ (Table A2). For AOD at 550 nm, we horizontally interpolate the products publicly available from each dataset to AERONET stations. The resolution of pNARA v1.0 is 0.5 by 0.5 degree; MERRA-2 is 0.5 by 0.625 degree; and CAMSRA is 0.5 by 0.5 degree. We also derived the AOD at 550 nm from companion AERONET AOD retrievals at other wavelengths for the comparison. Table A1 shows that the CAMSRA has smaller biases among the three datasets while pNARA v1.0 has smaller RMSE and larger $R^2$. Note that the overall $R^2$ for the whole dataset of AOD at 550 nm from pNARA v1.0, CAMSRA, and MERRA-2 are 0.768, 0.725, and 0.730, respectively. For $PM_{2.5}$, the derivation provided by GMAO (https://gmao.gsfc.nasa.gov/reanalysis/MERRA-2/FAQ/) is applied to MERRA-2 product. Similar to AOD at 550 nm, the horizontal interpolation is applied to MERRA-2 and CAMSRA, while it was done on native model grid for pNARA v1.0. Table A2 shows that CAMSRA overestimates the surface $PM_{2.5}$ while MERRA-2 and pNARA v1.0 both underestimate these observations. Like the comparison of AOD at 550 nm, pNARA v1.0 compares favorably in RMSE and $R^2$ to the other reanalyses with respect to the temporal variation of $PM_{2.5}$. These two

tables illustrate differences between the reanalyses. We note that a fairer comparison would result if all the reanalyses were available on native grids. Also, the overarching goal of this study was not to stratify various reanalyses in terms of accuracy. Instead, it was to show how significant are the differences that exist between them with respect to the mixing ratios of various aerosol species and the implications that this fact carries for modeling of impacts of aerosols on weather and climate.

**Code and data availability**

The GEFS-Aerosols and JEDI code we used to conduct pNARA v1.0 are public available on Zenodo (10.5281/zenodo.8226055). Because the size of reanalysis datasets is too large, we deposited the sample data of pNARA v1.0, MERRA-2 and CAMSRA on Zenodo (10.5281/zenodo.8222945). For pNARA v1.0, readers can browse the catalog of available files and retrieve the data via wget or curl commands based on the formats of url links below,
AOD file:
https://esrl.noaa.gov/gsd/thredds/fileServer/retro/global_aerosol_reanalysis/YYYYMM/NARA-1.0_AOD_YYYYMMDDHH.nc4,
Aerosol mass mixing ratio on model levels file:
https://esrl.noaa.gov/gsd/thredds/fileServer/retro/global_aerosol_reanalysis/YYYYMM/NARA-1.0_aero_YYYYMMDDHH.nc4,
where YYYY stands for the 4-digit year; MM stands for 2-digit month; DD stands for 2-digit day; HH stands for 2-digit hours. For instance, the fetching link of aerosol mass mixing ratio reanalysis on 12Z Aug 15, 2016 will be https://esrl.noaa.gov/gsd/thredds/fileServer/retro/global_aerosol_reanalysis/201608/NARA-1.0_aero_2016081512.nc4.
For MERRA-2, we used AOD (M2I3NXGAS) and aerosol mass mixing ratio (M2I3NVAER) datasets. It can be received by searching the tag in the parentheses on NASA's Goddard Earth Sciences Data and Information Services Center (GES DISC)
website (https://disc.gsfc.nasa.gov/). For CAMSRA, the data can be founded by 'EAC4' through Atmosphere Data Store website (https://ads.atmosphere.copernicus.eu/cdsapp#!/home). The CDS API is needed for users to fetch data (https://ads.atmosphere.copernicus.eu/api-how-to). The API request can be generated by selecting desired parameters on the website (https://ads.atmosphere.copernicus.eu/cdsapp#!/dataset/cams-global-reanalysis-eac4?tab=form) and users can retrieve files through Python script. The measurements from MODIS NNR, AERONET, OpenAQ, and IMPROVE for 2016 are
available on Zenodo (10.5281/zenodo.8226441).

**Author contribution**

Conceptualization and Investigation: MP, SW; Data curation, Supervision, and Project administration: MP; Formal analysis: SW, MP; Funding acquisition: MP, AdS, CL; Methodology: MP, BH; Resources: MP, AdS; Software: MP, BH, SW, AdS;

Validation and Visualization: MP, SW, BH; Writing – original draft preparation: SW, MP; Writing – review & editing: SW, MP, BH, CL

**Acknowledgements**

We are particularly grateful to Betsy Andrews from NOAA's Global Monitoring Laboratory for discussions and her drawing our attention to relevant research on radiative properties of aerosols and to David Giles for guidance on using AERONET retrievals. We very highly appreciate comments from Jérôme Barré that significantly improved the manuscript and acknowledge technical support of the whole JCSDA team. This research was funded by a grant from NOAA's Climate Prediction Office/Modeling, Analysis, Predictions and Projections, award number NA18OAR4310281 and by NOAA cooperative agreement NA22OAR4320151.

**Competing interests**

The authors declare that they have no conflict of interest.

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

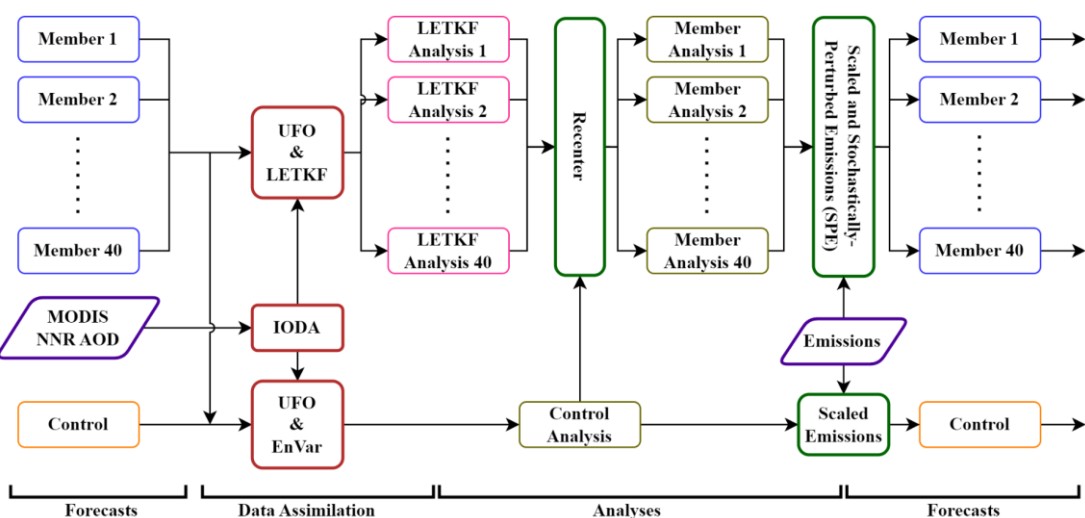

*Figure 1: Schematic of the 3D-EnVar data assimilation system for pNARA v1.0.*

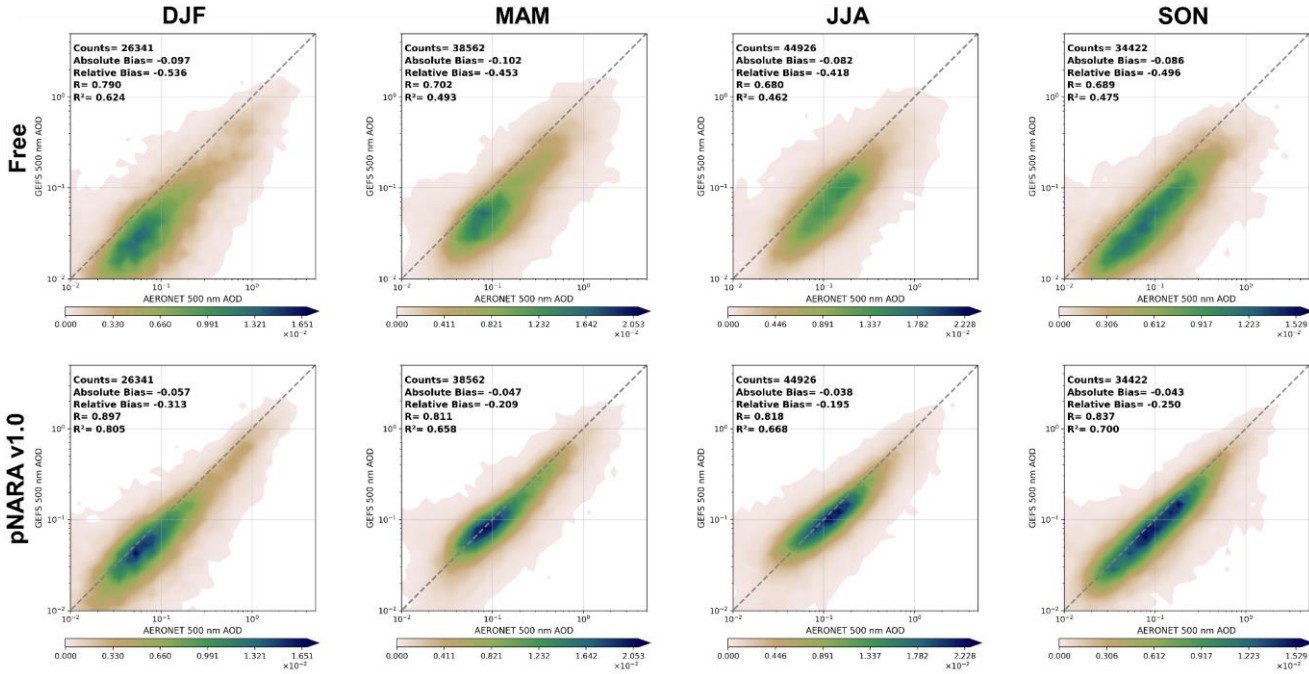

**Figure 2: Probability density plots of modelled AOD at 500 nm with respect to AERONET retrievals for the free model run and pNARA v1.0 during four seasons; axes are in logarithmic scale.**

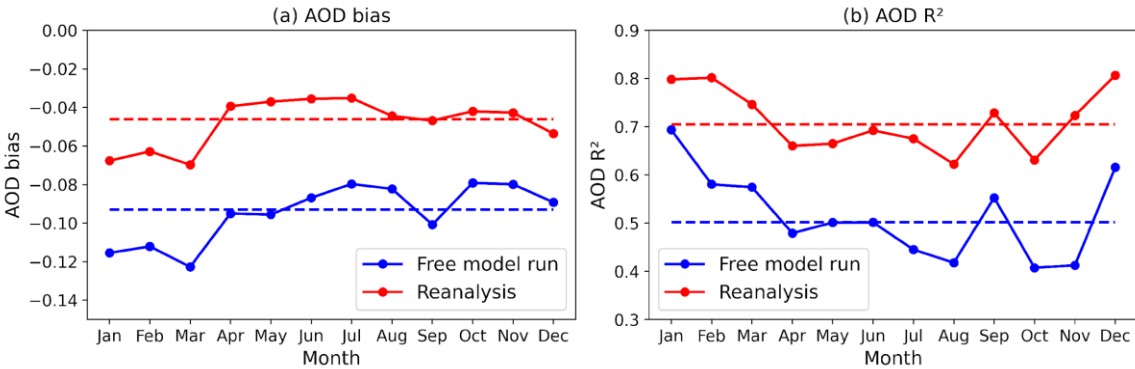

**Figure 3: The time series of the monthly mean of (a) biases and (b) $R^2$-correlation for the free model run (in blue) and pNARA v1.0 (in red) with respect to AOD at 500 nm measurements from AERONET. Both are calculated based on all the paired points within each month.**

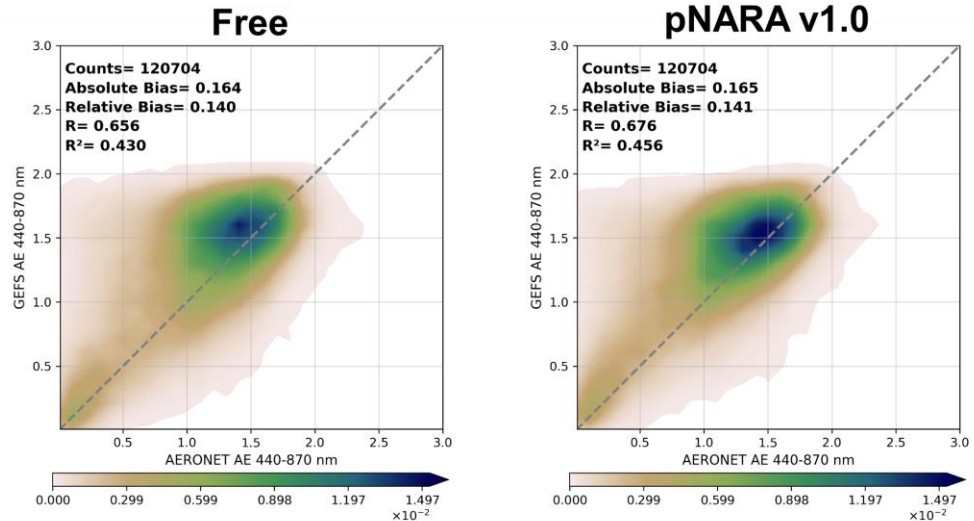

        **Figure 4: Probability density plots of 440-870 nm Ångström Exponent of the free model run (left) and pNARA v1.0 (right) vs.**
**AERONET for 2016.**

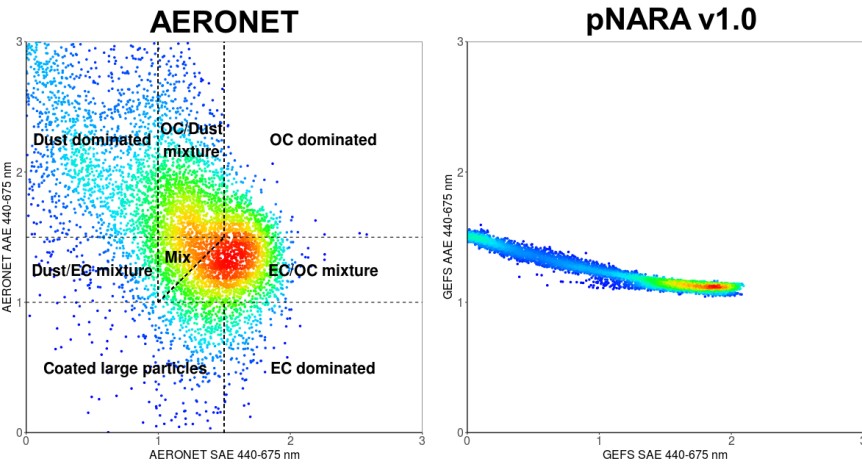

**Figure 5: Probability density plots of 440-675 nm Absorption Ångström Exponent vs. Scattering Ångström Exponent for AERONET (left) and pNARA v1.0 (right) for 2016.**

**Figure 6: The comparison of seasonal mean biases of AOD at 550 nm of pNARA v1.0 vs. CAMSRA and MERRA-2.**

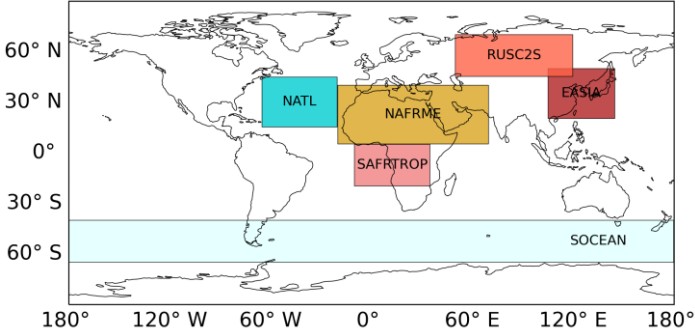

**Figure 7: The areas selected for comparison of vertical profiles of aerosols.**

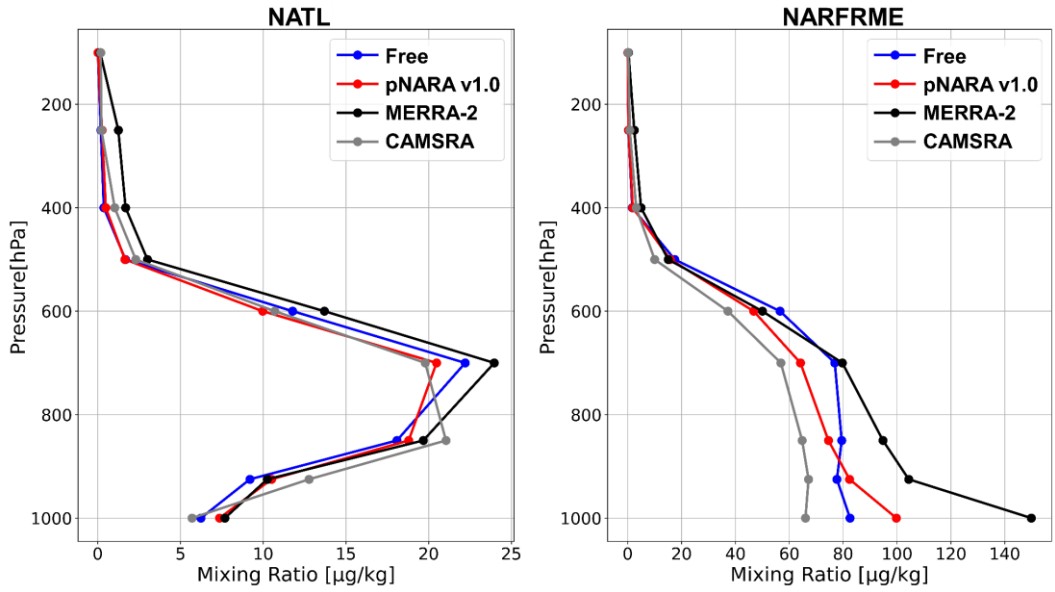

**Figure 8: Total dust mass mixing ratios profiles averaged over NATL (left) and NAFRME (right) during JJA.**

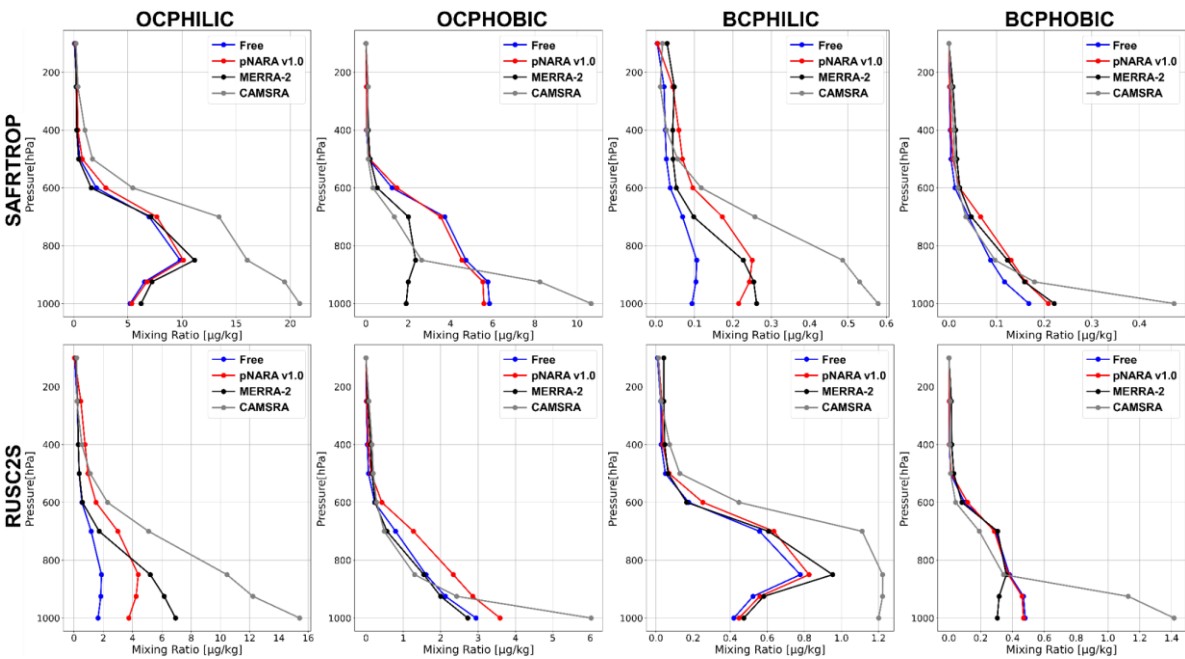


**Figure 9: Same as Figure 8, but for carbonaceous aerosols over SAFRTROP (top) and RUSC2S (bottom) during JJA.**

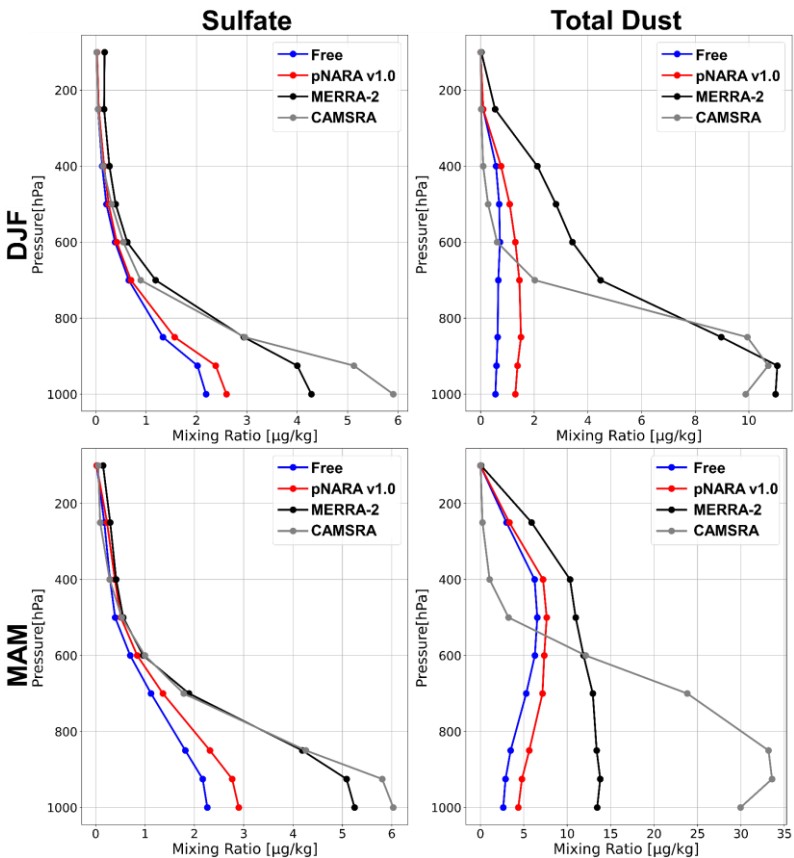

**Figure 10: Same as Figure 8, but for sulfate (left) and total dust (right) aerosols over EASIA during DJF (top) and MAM (bottom).**

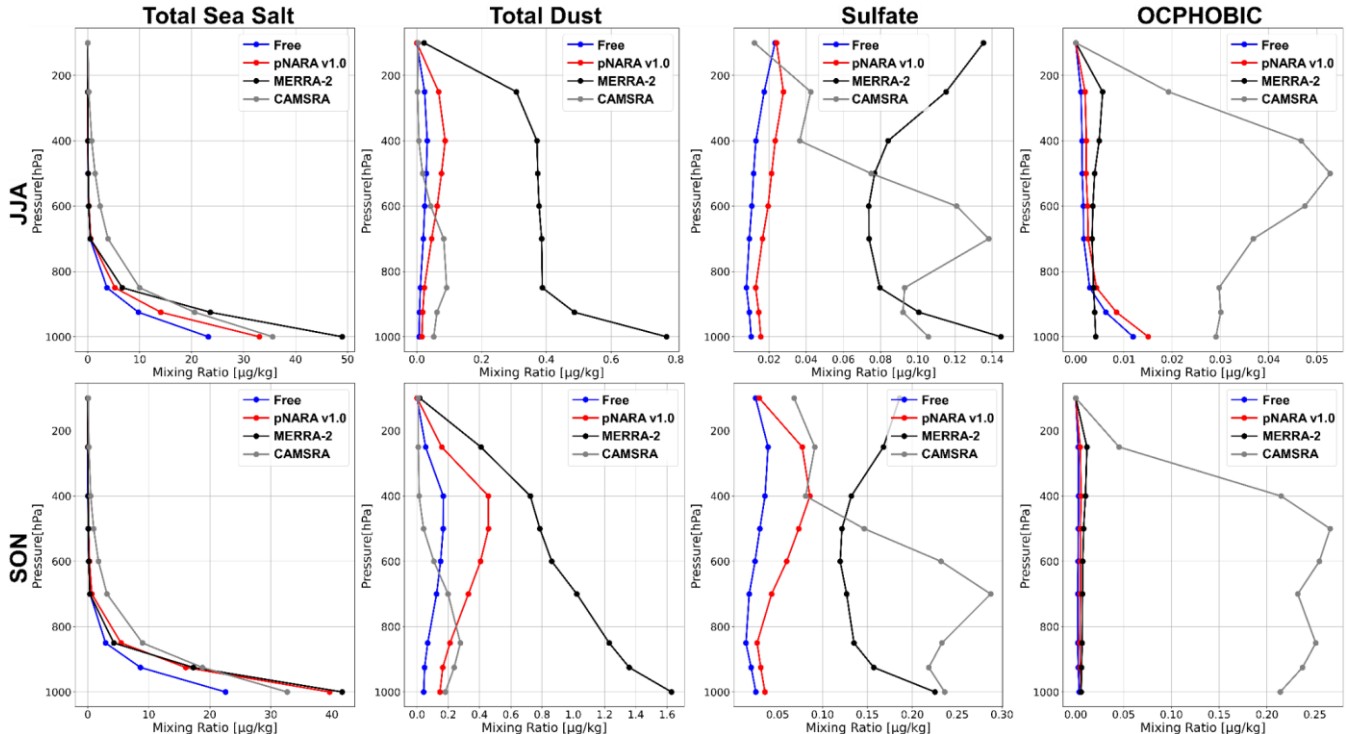

Figure 11. Same as Figure 8, but for sea salt, dust, sulfate, and hydrophobic OC over the Southern Ocean during JJA (top) and SON (bottom).

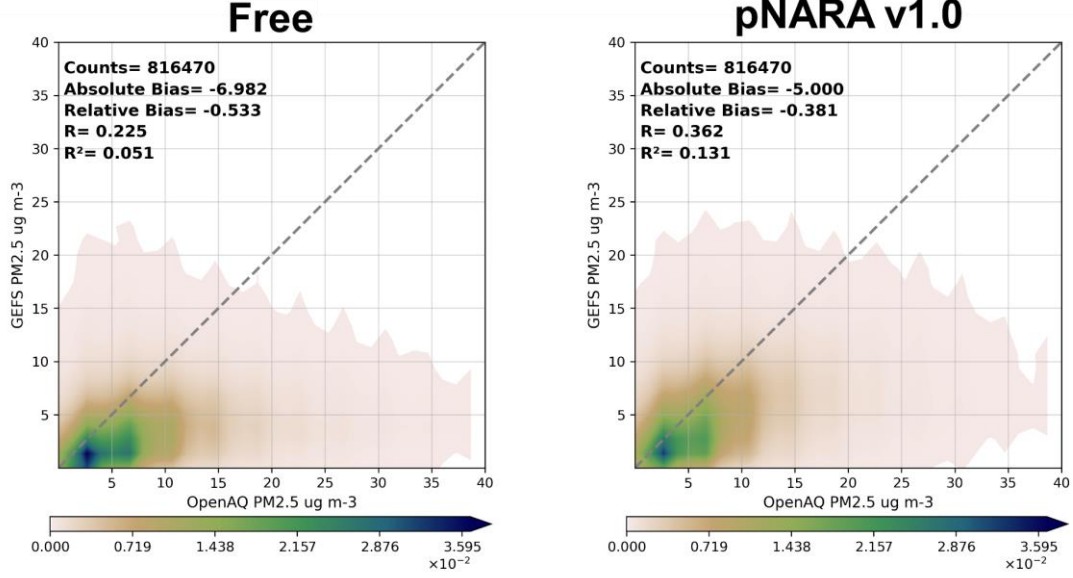

Figure 12. Probability density plots of modelled PM$_{2.5}$ with respect to OpenAQ measurements for the free model run (left) and pNARA v1.0 (right). Evaluation period: May – December 2016.

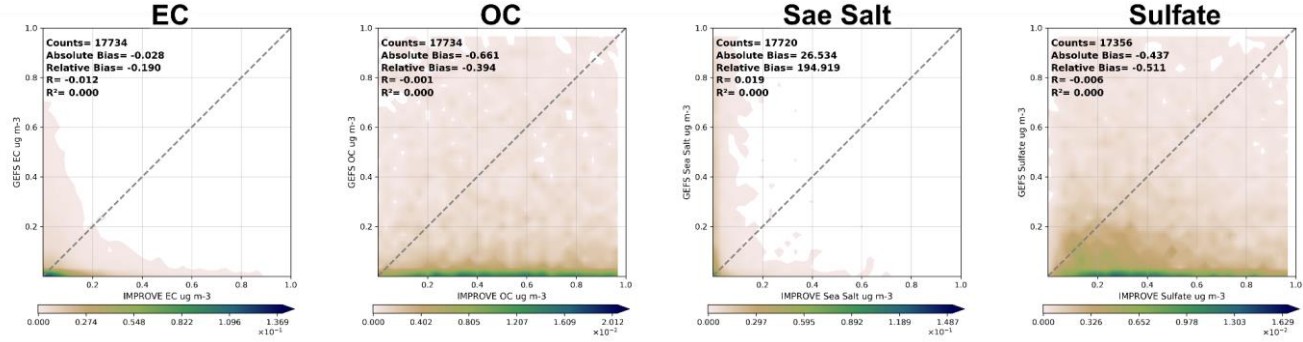


**Figure 13. Probability density plots of speciated aerosol surface concentration of pNARA v1.0 with respect to IMPROVE measurements for elemental carbon (EC), organic carbon (OC), sea salt, and sulfate (left to right).**

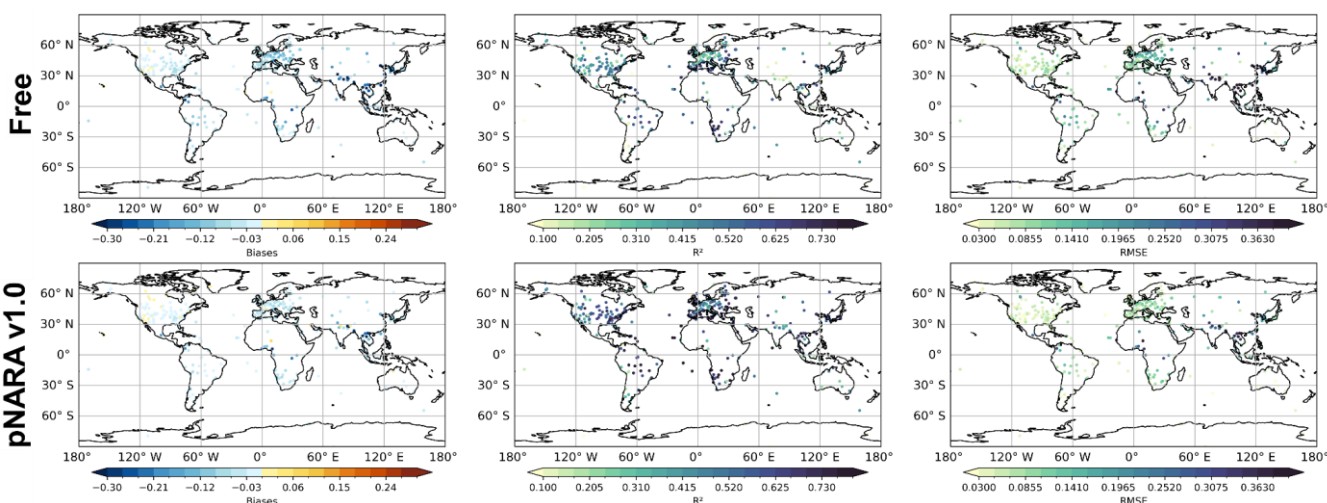

**Figure A1. The horizontal distribution of temporal biases (left), R²-correlation (middle), RMSE (right) of AOD at 500 nm against AERONET measurements of 2016.**

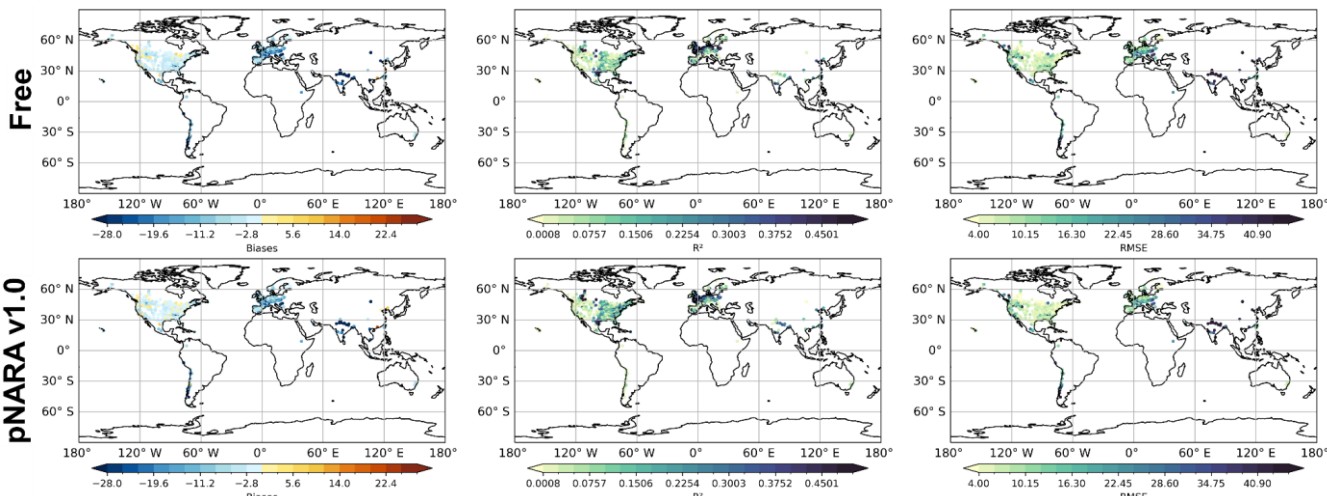

**Figure A2. The horizontal distribution of temporal biases (left), R²-correlation (middle), RMSE (right) of PM$_{2.5}$ (μg/m³) against OpenAQ measurements. Evaluation period: May – December 2016.**

**Table A1. The comparison of station averaged temporal statistics for bias, RMSE, and R²-correlation of pNARA v1.0, CAMSRA and MERRA-2 against AERONET AOD at 550 nm over four seasons and whole year 2016.**

|  | DJF | | | MAM | | | JJA | | | SON | | | Total | | |
|---|---|---|---|---|---|---|---|---|---|---|---|---|---|---|---|
|  | Bias | RMSE | $R^2$ | Bias | RMSE | $R^2$ | Bias | RMSE | $R^2$ | Bias | RMSE | $R^2$ | Bias | RMSE | $R^2$ |
| **pNARA v1.0** | -0.036 | **0.067** | **0.438** | -0.034 | **0.090** | **0.552** | -0.028 | **0.081** | **0.555** | -0.030 | **0.068** | **0.546** | -0.034 | **0.086** | **0.602** |
| **CAMSRA** | **-0.006** | 0.071 | 0.372 | **0.012** | 0.098 | 0.480 | 0.029 | 0.101 | 0.479 | **-0.001** | 0.075 | 0.492 | **0.008** | 0.094 | 0.526 |
| **MERRA-2** | -0.016 | 0.068 | 0.391 | -0.015 | 0.094 | 0.477 | **-0.013** | 0.087 | 0.492 | -0.015 | 0.073 | 0.474 | -0.015 | 0.089 | 0.540 |

**Table A2. The comparison of station averaged temporal statistics for bias, RMSE, and R²-correlation of pNARA v1.0, CAMSRA and MERRA-2 against OpenAQ PM$_{2.5}$ measurements over May to December 2016.**

|  | Bias (μg/m³) | RMSE (μg/m³) | $R^2$ |
|---|---|---|---|
| **pNARA v1.0** | -6.426 | **16.375** | **0.174** |
| **CAMSRA** | **3.167** | 19.583 | 0.134 |
| **MERRA-2** | -4.126 | 17.705 | 0.111 |
