# Peer review of "The Prototype NOAA Aerosol Reanalysis version 1.0 (pNARA v1.0)"

_EGUsphere, 2023_

## Author Comment (AC2)

Dear authors,

Unfortunately, after checking your manuscript, it has come to our attention that it does not comply with our "Code and Data Policy".

https://www.geoscientific-model-development.net/policies/code_and_data_policy.html

Your "Code and data availability" section reads:

JEDI code is available at https://github.com/JCSDA/fv3-bundle

GEFS-Aerosols code is available at https://github.com/SamuelTrahanNOAA/ufs-weather-model

NARA v1.0 data is available at https://esrl.noaa.gov/gsd/thredds/catalog/retro/global_aerosol_reanalysis/catalog.html

MERRA-2 data is available at https://disc.gsfc.nasa.gov/385

CAMSRA data is available at https://atmosphere.copernicus.eu/

OpenAQ data is available at https://openaq.org

IMPROVE data is available at http://vista.cira.colostate.edu/improve/

None of these repositories is accepted by our policy. What's more, GitHub is specifically mentioned as an unacceptable repository. GitHub is not a suitable repository for scientific publication. GitHub itself instructs authors to use other alternatives for long-term archival and publishing, such as Zenodo.

Therefore, please, publish your code and data in one of the appropriate repositories, and reply to this comment with the relevant information (link and DOI) as soon as possible, as it should be available for the Discussions stage. I should note that the license for the ufs-weather-model does not clarify what license applies to each part of the code in the GitHub repository. In this way, it is impossible to know what conditions apply to each part of the code. This problem should be addressed and solved.

About the data: MERRA2, NARA and CAMSRA data: It would be ideal if you could save the exact data that you have used in new files instead of simply pointing it out to generic download pages, where it can be hard to determine exactly the variable and data used in your work, precluding its replicability. Beyond the NOAA, NASA and

COPERNICUS servers, the openaq.org and Colorado State University repositories are not acceptable, and you must store the data in one of the acceptable repositories listed in our policy.

In this way, if you do not fix these issues, we will have to reject your manuscript for publication in our journal. I should note that, actually, your manuscript should not have been accepted in Discussions, given this lack of compliance with our policy. Therefore, the current situation with your manuscript is irregular.

Also, you remember including in a potentially reviewed version of your manuscript the modified 'Code and Data Availability' section, including the necessary DOIs.

Juan A. Añel
Geosci. Model Dev. Exec. Editor

**Response:**
**The Data and Code Availability section has been updated as below.**

The GEFS-Aerosols and JEDI code we used to conduct pNARA v1.0 are public available on Zenodo (10.5281/zenodo.8226055). Because the size of reanalysis datasets are too large, we deposited the sample data of pNARA v1.0, MERRA-2 and CAMSRA on Zenodo (10.5281/zenodo.8222945). For pNARA v1.0, readers can browse the catalog of available files and retrieve the data via wget or curl commands based on the formats of url links below,
AOD file:
https://esrl.noaa.gov/gsd/thredds/fileServer/retro/global_aerosol_reanalysis/YYYYMM/NARA-1.0_AOD_YYYYMMDDHH.nc4,
Aerosol mass mixing ratio on model levels file:
https://esrl.noaa.gov/gsd/thredds/fileServer/retro/global_aerosol_reanalysis/YYYYMM/NARA-1.0_aero_YYYYMMDDHH.nc4, where YYYY stands for the 4-digit year; MM stands for 2-digit month; DD stands for 2-digit day; HH stands for 2-digit hours. For instance, the fetching link of AOD reanalysis on 12Z Aug 15, 2016 will be
https://esrl.noaa.gov/gsd/thredds/fileServer/retro/global_aerosol_reanalysis/201608/NARA-1.0_aero_2016081512.nc4.
For MERRA-2, we used AOD (M2I3NXGAS) and aerosol mass mixing ratio (M2I3NVAER) datasets. It can be received by searching the tag in the parentheses on NASA's Goddard Earth Sciences Data and Information Services Center (GES DISC) website (https://disc.gsfc.nasa.gov/). For CAMSRA, the data can be founded by 'EAC4' through Atmosphere Data Store website (https://ads.atmosphere.copernicus.eu/cdsapp#!/home). The CDS API is needed for users to fetch data (https://ads.atmosphere.copernicus.eu/api-how-to). The API request

can be generated by selecting desired parameters on the website (https://ads.atmosphere.copernicus.eu/cdsapp#!/dataset/cams-global-reanalysis-eac4?tab=form) and users can retrieve files through Python script. The measurements from MODIS NNR, AERONET, OpenAQ, and IMPROVE for 2016 are available on Zenodo (10.5281/zenodo.8226441).

---

## Author Comment (AC3)

**Response to Reviewer #2 comments:**

**We thank reviewer #2 for the valuable comments and suggestions. According to the comments, we have revised the manuscript. Below we provide a point-to-point response to address the reviewer's comments.**

**Review of "The NOAA Aerosol Reanalysis version 1.0 (NARA v1.0): Description of the Modeling System and its Evaluation**

By Wei et al.

**Overview:** This manuscript describes an aerosol product generated for the year 2016, referred to as NARA v1.0, using the GEFS aerosol model and NNR AOD product assimilation using a 3Den-Var data assimilation. The results were compared to NASA's MERRA-2 and ECMWF's CAMSRA reanalysis products as well as against the AERONET AOD and surface PM2.5 data. The 2016 NARA v1.0 output was found to be more consistent with AERONET than the free model run. The results are also more consistent with MERRA2 than CAMSRA reanalysis. This isn't really surprising as the NARA v1.0 setup has more similarities to MERRA-2, using GOCART aerosol and NNR AOD product that is also assimilated for MERRA-2.

**General Remarks:** The main issue I have with this manuscript is referring to the output as a reanalysis when the results are only shown for 1 year. A reanalysis is typically over a long period of time (10years+) as is the case for the other aerosol reanalysis products that are available. I would call this an evaluation paper of the performance of the 3Den-Var with the GEFS model. I think it's fair to say that you will apply this setup in the future for generation of a reanalysis product, but don't think you can call it that here. Given that, I think more needs to be done to define what is different here than in the Huang et al. 2023 paper that defined the data assimilation setup and evaluates the analysis results, although for only a month time period versus a year. I also think there are more details needed in the manuscript, including the data assimilation setup. More description of observations and reanalysis data used in this paper would also be helpful. For the observations, was there any QC done prior to using the data? Was there any temporal averaging done in order to make point data comparable to model output? I do want to highlight that reanalysis products are very important and once you are able to generate that, I think that will be a valuable contribution.

**Response: We agree the length of dataset is not sufficient for a reanalysis dataset, so we have renamed this 2016 reanalysis product as the prototype NOAA Aerosol ReAnalysis version 1 (pNARA v1.0). For the DA system, we have revised the description, especially the detail of use of MODIS NNR AOD. Specifically,**

**when the AOD is retrieved via neural network, the cloud-affected data points were screened out and we also thinned the resolution from 10 km to 50 km. We added the pre-process procedure of reanalyses dataset in the beginning of Section 5.2. Please see below for point-to-point response for specific comments.**

**Specific Comments:**

- First sentence in the abstract. I see what you are saying here, but I think this could be a bit misleading. It could come off as you have the first aerosol reanalysis product ever, which is not the case. Please reword this as "the first version of the aerosol reanalysis for NOAA", or something like that in order to clarify.

  **Response: The first sentence has been revised according to reviewer's comment. Now it is 'the first prototype version of the aerosol reanalysis...' (L1).**

- Page 4, first line: "In this study, we used and designed a specific JEDI-based 3D-EnVar DA configuration to produce the NOAA Aerosol ReAnalysis version 1.0 (NARA v1.0). " Please elaborate on how this differs from the setup described in Huang et al. 2023. The difference is not clear and why is the chosen configuration better for a reanalysis?

  **Response: It is the same DA system as Huang et al. (2023), but the MODIS NNR AOD at 550 nm is assimilated in pNARA v1.0 and the configurations of scale factor and perturbation of emission are adjusted accordingly.**

- Have you also looked at performance of fine mode fraction or fine/coarse AOD?

  **Response: No, the fine mode fraction or fine/coarse AOD were not assessed in this study, because these information is not available in MODIS NNR AOD retrievals. This task will be pursued in the future when we introduce the assimilation of multiwavelength retrievals/measurements into the system.**

- I think looking at timeseries that are not monthly-averaged would also be helpful, at AERONET sites for example, to see if your product is able to generate daily variability. Perhaps you can select reference sites that have a good amount of data and at locations that are representative of big dust/smoke/pollution/sea salt dominated regimes.

**Response: Given the horizontal resolution of pNARA v1.0 is in 1 degree, we mainly focused on the performance at large spatial and temporal scale. We have added discussion and figures of temporal statistics against AERONET in Appendix A.**

---

## Author Comment (AC4)

**Response to Reviewer #1 comments:**

**We thank Reviewer #1 for the valuable comments and suggestions. According to the comments, we have revised the manuscript. Below we provide a point-to-point response to address the reviewer's comments.**

Review of

The NOAA Aerosol Reanalysis version 1.0 (NARA v1.0): Description of the Modeling System and its Evaluation

By Wei et al.

**Overview:**

The paper presents a global aerosol reanalysis data set for the year 2016 (NARA v1.0) produced by assimilating MODIS AOD in the GEFS aerosol model using the JEDI data assimilation framework. The reanalysis data and a control run (free) without AOD assimilation is evaluated with AOD and Angstroem exponent (AE) observations from Aeronet, surface PM observations derived from openAQ and speciated observations from the IMPROVE network. Further, it is intercompared against the MERRA-2 and CAMS aerosol reanalysis data sets. The authors find improved AOD compared to AERONET observations but no or only smaller improvements for AE and PM. Overall NARA v1.0 seems closer to the MERRA-2 re-analysis than to CAMS.

**General remarks**

The paper gives a reasonable description of the modelling system and the scientific accuracy of the NARA data set. However, there are further updates required before the paper can be recommended for publication.

- It is an omission of the paper that NARA, CAMS and MERRA-2 are only compared against each other for AOD and regional vertical profiles. Instead, the evaluation against independent observations should be carried for CAMS and MERRA-2 too. The accuracy measure of the three data should be compared in the paper. It is of great interest to the reader to get know which of these data sets is closer to observations.

- The evaluation results should be represented in a more quantitative manner and always for NARA and the FREE run to document the impact of the AOD assimilation. For example, the pdf plots do not have a colour scale legend and only some of them contain information about bias and correlation.

- The approach for the PM evaluation needs to be described in more detail, in particular the seemingly missing application of any quality control of the openAQ data should be justified. Further, the method to account for vertical stratification and spatial representativeness is not always clear.
- The abstract should be revised to contain more factual information from the paper. General remarks that are as such not direct conclusion from the paper should be avoided.
- The data assimilation procedure seems to include an optimisation of the emissions (see also Fig 1) Please provide more detail on this important aspect from a technical and a scientific perspective.

**Response:**

**Thanks for the comments. We have revised the manuscript to address the comments. Please see the point-to-point response below.**

**Specific comments:**

L 11: once published that is not a manuscript

**Response: It has been revised to 'study' (L1).**

L 11: "first ever" may be misleading because there are other aerosol re-analysis

**Response: The "first ever" has been revised to "first prototype version of" (L1).**

L 24: There is no clear evidence in the paper that single-wavelength assimilation is the main factor of the limited impact.

**Response: The interpretation statement (next sentence) has been removed and the relevant discussion in section 5 has been revised accordingly (L209-213).**

L 28: What is meant here by "climatologies"

**Response: It has been modified to 'among datasets' (L27).**

L 28: "In our opinion, such uncertainties may translate to inaccuracies in weather and climate modeling when impacts of aerosols on atmospheric radiation and/or cloud processes are considered." The paper does not deal in any way with the impact of aerosol and radiation and cloud processes and it is therefore not a conclusion of this paper. Please avoid statements that are not substantiated by the paper.

**Response: This sentence has been removed in the revision and the relevant discussion in Section 6 is revised (L370-373).**

L 71: Please check of SOA's and Nitrates are included in CAMSRA (I believe not)

**Response: We confirmed that SOA is included in CAMSRA, which is detailed in section 2.1.1 of Inness et al. (2019). This is not considered in the previous version, CAMS interim reanalysis, which is the 'CIRA' in Iness et al. (2019) and 'CAMSiRA' in our study.**

L 114: openAQ data are know for the lack of QC. Please expand on that and clarify the data sources that were compiled in the openAQ data set for 2016. A good solution would be to provide a global map to show the spatial distribution of the PM2.5 observations.

**Response: To evaluate pNARA v1.0 against the OpenAQ dataset, we only utilized measurements from reference grade monitors (i.e. those maintained by government/research institutes) which are considered "the gold standard" (https://openaq.org/developers/help/). This detail has been added in the revision (L121-125).**

L 123: Why are gases mentioned here? Did you use the data , for example SO2 ?

**Response: Additional information is helpful for future utilization of the dataset. The sentence has been revised accordingly (L124-125).**

L 134: Comment on the importance of representing Nitrates and Secondary Organic Aerosol especially for the accuracy of PM.

**Response: SOA and nitrates may constitute significant parts of fine PM depending on season and geographical location. Since the formation of SOA involves condensation of gases and complex chemical reactions including the organosulfates, organic nitrates, and interactions of ozone with free radicals (OH, NO3) computational expense of including its parameterization in a global model like ours is beyond consideration. We expect that impact of neglecting SOA in PM would be largest over urban areas. Inorganic nitrates are indeed included in a recent extension of GOCART at NASA, GOCART2G, and will be accounted for in calculating PM in the future. We expect that the impact of neglecting nitrates in PM would be largest over agricultural areas. Rather than speculate we will not attempt to estimate by how much our results are influenced by lack of accounting for SOA and nitrates.**

L 170-174: Please expand on these aspects (SPPT and emission updates) and provide more information about their usefulness for the realism of the NARA data set.

**Response: The description has been expanded with how the scale factor and perturbation of emission can impact the analyses in pNARA v1.0 in the revision (L173-179).**

L 174: Please show and discuss the modified emissions.

**Response: The discussion of the impacts of scaling and perturbation to emissions on reanalyses is beyond the scope of the study. The influence of modified emissions induced by the scaling factor and perturbation are detailed in Huang et al. (2023).**

L 205: Please provide more evidence for this statement "We hope ...."

**Response: The statement has been revised and citations for the benefits/improvements induced by assimilating multiwavelength observations were added (L209-213).**

L 232: Section 5.2 should also include a verification of CAMS-RA and MERRA-2 consistent with the verification of NARA and a comparison of the results

**Response: We agree the intercomparison between reanalysis datasets may contain important information for users. However, it is beyond the scope of this study because we make no representation of other agencies. We believe that a fair intercomparison study would require a broad collaboration with others. Therefore, the citation of the evaluation papers of MERRA-2 and CAMSRA have been added for reader's reference in Section 6 (L363).**

L 304: Please, clarify how you account for the stratification. This would require making assumptions about the vertical aerosol profiles within the model grid box. Just using the diagnosed air density, will not achieve that.

**Response:**

**We agree that deriving surface concentrations of species would require (numerous) assumptions. Below we elaborate why we chose a simpler approach.**

**As noted in the manuscript, the model mid-level where mixing ratios of species are calculated is at 20 m. This in most cases would fall within the atmospheric**

**surface layer. Vertical profiles of concentrations/mixing ratios of fine aerosols in the surface layer follow Monin-Obukhov theory and are expressed in the form**

**$c - c_0 = c*/k \, [\ln(z/z_c) + \psi(z, z_c, L)]$, where**

**$c$ is species concentration at layer depth $z$, $c_0$ is species concentration at the top of the interfacial layer, $z_c$ is species dependent height scale, $\psi$ is stability dependent function, $L$ is Monin-Obukhov length, $k$ is von Karman constant, and $c*$ is surface flux divided by friction velocity (e.g. The Atmospheric Boundary Layer, Garratt, J., 1992; Atmospheric Chemistry and Physics: From Air Pollution to Climate Change, Seinfeld, J.H, Pandis, S.N, 2016).**

**From this equation a formula for a concentration of species at measurement level can be derived by plugging the known concentration at model level and eliminating $c*$. That leaves us with $c$ at the measurement level as a function of stability dependent $L$ and poorly characterized $z_c$ and $c_0$. We could simplify the equation by assuming neutral stratification (i.e. $\psi = 0$) but $z_c$ and crucially $c_0$ would need to be further parameterized. We don't believe that following this theory-based approach would provide a better approximation to the one we adopted. Our approach accounts solely for air compression i.e. by scaling model level concentration using densities at the lowest model level and at the measurement level assumed to be 2m (though it is also quite uncertain at what height the actual OpenAQ measurements are taken). We performed the evaluation of the modeled PM2.5 being fully aware of the shortcomings of such an effort.**

L 307: Please provide information about the size distribution of the different aerosol component and motivate the PM formulae.

**Response: The formula was provided by the lead author of the publication on the GEFS-Aerosol model (Geosci. Model Dev., 15, 5337–5369, https://doi.org/10.5194/gmd-15-5337-2022, 2022, Kate Zhang, personal communication, March 20, 2023). We believe that the formula is derived from the prescribed distributions of the mass of the aerosol component species given their mean radii and standard deviations.**

L 358: Please add a comparison of the evaluation results from MERRA-2, CAMS and NARA here.

**Response: Please see the response for L232.**

L 362-365: "These observations … " That is a very general statement and not really justified by the findings of the paper as you do not test the impact of NARA aerosol on

radiation and clouds. Please include multiple references if you want to make a point here. For example, you mention yourself that Bozzo et al. 2017 (there is also an ACP paper) successfully used CAMSRA to represent aerosol in the ECMWF model.

**Response: Bozzo et al. (2017) and Mulcahy et al. (2014) have been cited in the revision to point out the importance of including the more representative aerosol states in the physical processes of aerosol-cloud-radiation interactions.**

Figures:

Fig 1:. Please here or in the text, provide more details in particular on the emission updates, the observation operators and the length of the forecast and assimilation window.

**Response: The information regarding the system configuration has been revised in Section 4.**

Fig 2:  Please add colour legend and basic statistics (bias, R) . Please show AOD in linear scale and not in log scale.  The pdf of AOD does not justify the use of the log scale.

**Response: The color bars and statistics have been added to each sub-panel and thus Table 2 is removed in the revision. The logarithmic scale is used to increase the visibility of the agreement between model and baseline for small values of AOD, which was utilized in the evaluation of MERRA-2 (Buchard et al., 2017).**

Fig 3:  Please, clarify if you show a spatial or temporal Correlation coefficient R.

**Response: This correlation coefficient is calculated based on the all paired data within each month. The figure caption has been revised accordingly.**

Fig 12: Please add a colour legend, consider showing the plot for different regions and not just a global plot.

**Response: The colorbar and statistics have been added. We also added the 2D map of temporal mean biases, RMSE, and correlation coefficient in Appendix A for reference. As shown in Figure A2 in the revision, the 'reference grade' stations in the OpenAQ dataset are mainly located over US and Europe. It does not show discernible improvements in pNARA v1.0. Compared to smaller biases over US and Europe, the substantial underestimates and large RMSE over India indicate the GOCART scheme cannot adequately capture the processes.**

---

## Author Response (AR2)

Reviewer #1 comment

The paper has been improved by responding to my technical comments in a reasonable way. However, the main omissions, namely the need to also present the AOD-AERONET and PM2.5 openAQ evaluation for CAMSRA and MERRA-2 has not been addressed. It is unsatisfactory that a paper which shows in detail differences between the NARA, CAMSRA and MERRA-2 (Figs. 6, 7-11) limits the evaluation with observations (Fig. 2, 12, A1) to NARA. The presented evaluation is relatively simple and repeating it for MERRA-2 and CAMSRA seems doable, and it is of great value.

**Response: Thanks for the interest in the intercomparison between pNARA v1.0 and other reanalysis datasets and we agree it is valuable information. As we mentioned, however, the evaluation of three or more datasets requires collaboration of agencies to have the agreement on the approach, and this is not the main purpose of our study. We would like to keep our focus on substantial differences among datasets which reflect the state of art of aerosol science rather than the evaluation of performance from multiple reanalyses. Hence, we added the further comparison among reanalyses with respect to AOD at 550 nm from AERONET and surface $PM_{2.5}$ from OpenAQ in Appendix A for reader's information.**